# Neem and Gliricidia Plant Leaf Extracts Improve Yield and Quality of Leaf Mustard by Managing Insect Pests’ Abundance Without Harming Beneficial Insects and Some Sensory Attributes

**DOI:** 10.3390/insects16020156

**Published:** 2025-02-03

**Authors:** Rowland Maganizo Kamanga, Salifu Bhikha, Felix Dalitso Kamala, Vincent Mgoli Mwale, Yolice Tembo, Patrick Alois Ndakidemi

**Affiliations:** 1Department of Horticulture, Lilongwe University of Agriculture and Natural Resources, Lilongwe P.O. Box 219, Malawi; vmgoli@luanar.ac.mw; 2Center of Excellence in Transformatice Agricultural Commercialization and Entrepreneurship (TACE), Lilongwe P.O. Box 219, Malawi; 3Department of Forestry, Lilongwe University of Agriculture and Natural Resources, Lilongwe P.O. Box 219, Malawi; bhikhasalifu@gmail.com (S.B.); fkamala@luanar.ac.mw (F.D.K.); 4Department of Crop and Soil Sciences, Lilongwe University of Agriculture and Natural Resources, Lilongwe P.O. Box 219, Malawi; ytembo@luanar.ac.mw; 5School of Life Sciences and Bio-Engineering, Nelson Mandela African Institution of Science and Technology, Arusha P.O. Box 447, Tanzania; patrick.ndakidemi@nm-aist.ac.tz

**Keywords:** botanical extracts, neem, gliricidia, integrated pest management, beneficial insects, sensory attributes

## Abstract

**Simple Summary:**

Over time, the demand to increase food production in order to feed the world’s increasing population has led to the excessive use of agrochemicals such as pesticides, putting human health and essential ecosystem services at risk. In this study, we explored plant-based extracts in managing insect pest infestation in leaf mustard. We report that by lowering the number of insect pests and preserving an optimal level of beneficial insects, neem and gliricidia extracts improved the yield and quality of leaf mustard. Moreover, unlike synthetic pesticides, the botanical extracts did not significantly alter sensory attributes. Hence, these plant extracts represent promising pesticidal plant materials and botanically active substances that can be leveraged to create commercial pest management solutions that are safe for the environment and human health.

**Abstract:**

Production and consumption of vegetable crops has seen a sharp increase in the recent past owing to an increasing recognition of their nutraceutical benefits. In tandem, there has been unwarranted application of agrochemicals such as insecticides to enhance productivity and vegetable quality, at the cost of human health, and fundamental environmental and ecosystem functions and services. This study was conducted to evaluate the efficacy of neem and gliricidia botanical extracts in managing harmful insect pest populations in leaf mustard. Our results report that neem and gliricidia plant extracts enhance the yield and quality of leaf mustard by reducing the prevalence and feeding activity of harmful insect pests in a manner similar to synthetic insecticides. Some of the key insect pests reduced were *Lipaphis erysimi*, *Pieris oleracea*, *Phyllotreta Cruciferae*, *Melanoplus sanguinipes*, and *Murgantia histrionica*. However, compared to synthetic insecticides, neem and gliricidia plant extracts were able to preserve beneficial insects such as the *Coccinellidae* spp., *Trichogramma minutum*, *Araneae* spp., *Lepidoptera* spp., and *Blattodea* spp. Furthermore, plant extracts did not significantly alter sensory attributes, especially taste and odor, whereas the visual appearance of leaf mustard was greater in plants sprayed with neem and synthetic insecticides. Physiologically, plant extracts were also able to significantly lower leaf membrane damage as shown through the electrolyte leakage assay. Therefore, these plant extracts represent promising pesticidal plant materials and botanically active substances that can be leveraged to develop environmentally friendly commercial pest management products.

## 1. Introduction

Vegetable crop production is a predominant source of livelihood in Malawi, especially among female-headed households. With the changes in consumption patterns favoring the consumption of vegetable crops in light of increased awareness of their nutraceutical and health benefits, the production of vegetables has seen a tremendous increase in the recent past. Leaf mustard (*Brassica juncea*) is among the most commonly cultivated vegetable crops in Malawi [1]. However, the production of leaf mustard is often challenging, particularly during the rainy season, due to various biotic constraints. Arguably, the most important biological constraint to vegetable crop productivity for smallholder farmers is high infestation by insect pests, inevitably necessitating the application of synthetic pesticides [2], which are extremely hazardous to human health and perturbs ecosystem balances.

In light of increased awareness of healthy consumption patterns and environmental stewardship, it is imperative to explore options to manage insect pest populations in ways that are less hazardous to human health and that maintain vital ecosystem services. Presently, many candidate plant species exist with known pesticidal properties, in terms of chemistry and efficacy under laboratory conditions, that could potentially be leveraged into developing new products [3]. Hundreds of indigenous and exotic species with pesticidal properties have been reported from Malawi through various farmer surveys and subsequent research, many of which have been confirmed to be active against a range of arthropod pests [4,5]. Moreover, on-farm use of pesticidal plants, particularly among resource-poor smallholder farmers, is widespread and remains a familiar practice to many Malawian farmers. These plant extracts are more advantageous over synthetic pesticides, as they prevent the development of insecticide resistance due to the presence of several bioactive compounds, have low persistence in the environment, and have a low cost of use for smallholder farmers [6,7] (Angioni et al., 2005; Gopalakrishnan et al., 2014). Major setbacks for plant extracts include variable efficacy, low toxicity, and persistence against target pests due to the rapid breakdown of bioactive compounds, e.g., through photodegradation and ease of wash due to rains [8].

According to Cadsawan et al. (2020), it has been reported that over 2000 plant species possess insecticidal properties. For instance, some plants produce a distinctive odor or chemicals that are repulsive to insects and others show ovipositional inhibitory activity. Some of the most commonly reported plant extracts with insecticidal effects include *Gliricidia sepium* (gliricidia) and *Azadirachta indica* (neem)*. Gliricidia sepium* is a leguminous tree that belongs to the family of *Fabaceae*. The tree species is known as a multipurpose tree that is commonly used in agroforestry and apiculture. The generic epithet gliricidia means mouse killer as its seed, bark, and leaves are said to contain toxic substances that kill not only mice but also insect pests [9]; hence, these properties could be leveraged as botanical pesticides. In addition, *Azadirachta indica* (neem) belongs to the family of *Meliaceae* and is known to possess insecticidal activity against more than 350 insects [10]. Hence, it can be used to manage various pests that attack agricultural crops including its extracts being used in managing fall army worms (FAW) infestation for maize in Malawi [4].

This study was conducted to evaluate the efficacy of plant extracts in managing harmful insect pest populations in leaf mustard under open-field conditions. Furthermore, their role in preserving beneficial insects in comparison with synthetic pesticides was explored. We demonstrate that neem and gliricidia leaf extracts enhance the yield and quality of leaf mustard by reducing the prevalence and feeding activity of harmful insect pests, while also preserving beneficial insects. These plant extracts represent promising pesticidal plant materials and botanically active substances that can be leveraged to develop environmentally friendly commercial pest management products.

## 2. Materials and Methods

### 2.1. Study Site

The study was an open field trial conducted at field sites in Malawi during the 2022/2023 rainy season. Malawi is found in the Southeastern part of Africa and is characterized by subtropical conditions with a subhumid climate. Field trials were carried out at the Horticultural farm of the Lilongwe University of Agriculture and Natural Resources (LUANAR), Bunda campus, Lilongwe, Malawi. Lilongwe district lies in the mid-altitude. The farm is situated at 14°35′ S, 33°50′ E, 1158 m above sea level. The site receives an average rainfall of 1030 mm/year with an average temperature of 20 °C and a mean maximum temperature of 29 °C. The growth environment had an average humidity of 80%. The soil at the site was classified as alfisol according to USDA (2022). The site is a frequently cultivated field with minimal vegetative cover.

### 2.2. Preparation of Plant Extracts

The botanical pesticides were prepared from leaf extracts of *G. sepium* and *A. indica* species using the water extraction method. These plant species were chosen due to their wide abundance around farms, roadsides, and bushland; their familiarity to farmers; and considerable existing knowledge of their efficacy, bioactive constituents, and safety. The leaves of all plant species were collected from the wild in Mitundu, Lilongwe. Extraction was made according to a method earlier described in Phambala et al. (2020). For this purpose, plant materials were shed dried, ground to a fine powder, and kept in cool dark conditions until required. To produce 10% *w*/*v* extracts, 100 g of each plant powder was filled to 1 L of water containing 0.1% detergent soap and extracted at room temperature (20 °C ± 5) for 24 h. A total of 0.1% detergent was added to increase the extraction efficiency of non-polar compounds from plant material. Thereafter, the extracts were filtered and used immediately in bioassays. The trial consisted of two treatments (neem and Gliricidia plant extracts), a positive control (synthetic pesticide), and a negative control (water plus 0.1% soap). Thereafter, the extracts, together with the two controls were poured into separate 15 L knapsack sprayers and used in the field. The application was done during the late afternoon between 16:00 to 17:00 h to avoid the hot sun from biodegrading the active compounds. The application was done at a concentration of 10% *w*/*v* as this was determined to be the most effective concentration for reducing insect damage and maintaining high crop yield according to previous studies [4,11]. Plant extracts were applied every 4 days because of frequent washaways and high pest abundance resulting from frequent rains.

### 2.3. Plant Materials and Experimental Design

Prior to planting, the farmland was deep-tilled and then ridged. Ridges were made at a spacing of 75 cm apart. Leaf mustard (*Brassica juncea*) was cultivated following traditional practices, where the seed is sown on nursery beds, hardened off, and transplanted to the ridges after attaining pencil-size thickness (21 days after sowing). Leaf mustard cultivar Florida broad leaf was used, sourced from Starke Ayres Seed Company, Capetown, South Africa. On ridges, the seedlings were planted at a spacing of 75 cm between rows and 30 cm within rows. Plot sizes were 5 m by 5 m, leaving a 1 m distance between each plot. The plots and treatments were arranged in a randomized complete block design (RCBD) with four blocks (Figure 1).

### 2.4. Insect Pest Occurrence and Plant Damage

The insect pest occurrence was done on a weekly basis using visual counts. Trends of insect pest population were observed from two inner rows of the net plot. Five plants were selected from the two middle rows for visual examination to record the number of each insect type. Approximately 5 min were spent on the examination and identification of each plant, more time was spent on a plant with a newly identified insect. For insect identification, insects were identified at the guild/functional group level according to a table developed by Anord et al. (2021) [12]. For insect identification, the data collectors focused on more obvious life stages and relatively larger insects for easy identification. If identity was still uncertain, insect samples were collected and preserved in 70% ethanol for further identification in the laboratory using fact sheets and taxonomic keys for arthropod identification. Insect pest occurrence was done by observing the number of insect pests available a day after the application of the pesticides and botanical extracts. The main insect pests under observation were flea beetles, caterpillars, aphids, grasshoppers, and red bugs due to their abundance in occurrence and economic importance in cabbage. The number of plants affected after the application of the pesticides was collected by counting the number of plants showing insect pest presence in the net plot of each treatment condition. In order to account for environmental variability (e.g., microclimate differences, pest pressure) across the plots, the field was laid out in a randomized complete block design (RCBD), and the observations were done in all four blocks. Each block constituted all the treatment conditions and accounted for the plausible variations in the microclimate.

### 2.5. Occurrence of Beneficial Insects

In order to evaluate the effect of plant extracts on the occurrence of beneficial insects, the number of non-targeted beneficial insects was recorded from all treatments across the blocks. The beneficial insects identified and monitored were roaches, small butterflies, predatory beetles, wasps, and spiders.

### 2.6. Biological and Economic Yield

The biological yield was obtained as the total dry matter produced per plant. It included all of the leaf, stem, and root dry matter produced by the plant. This was measured by harvesting 10 plants from each plot and weighing each separately on a balance including pest-infested leaves and obtaining mean values. Economic yield was obtained by weighing only healthy leaves usable for human consumption.

### 2.7. Total Leaf Area

The total leaf area was measured from the leaf blades of 10 biological replicates from each treatment in each block using ImageJ software version 1.54e. Leaf area was taken after removing all infested leaves and so only represented economic leaves.

### 2.8. Membrane Integrity Parameters

Membrane integrity was assayed using shoot electrolyte leakage (SEL) assay according to Kamanga et al. (2023) [13]. For this purpose, plant leaves with visible insect damage were sampled from each treatment and cut into smaller uniform-sized disks about 2.5 cm wide. Thereafter, the cut leaf disks were dipped into 15 mL falcon tubes containing 12 mL distilled water and kept at 10 °C for 24 h. Thereafter, the samples were warmed at 25 °C. Electrical conductivity (EC) was then obtained (EC1) using an EC meter (Model AZ 8303, AZ Instrument Corporation, Taichung, Taiwan). Immediately, the samples were killed by autoclaving at 105 °C for 15 min and EC2 was obtained. SEL was then obtained as follows:SEL=(EC1/EC2)×100

### 2.9. Sensory Evaluation

This evaluation was conducted in order to determine whether the plant extracts and synthetic insecticides applied influenced sensory attributes and consumer acceptability of the leaf mustard. This was conducted by asking respondents to rate mustard leaves from 1 to 10 accompanied by a visual representation adapted from Munkhuwa et al. (2023) [14] based on sensory characteristics of the mustard, namely, taste, appearance, and odor. A two-week safety period was observed before harvesting from the previous application of pesticides. A total of 43 participants were involved in the sensory analysis, of which 15 were trained food science and technology students on sensory evaluation. All the participants were randomly selected from their clusters (trained and untrained). In order to avoid biases, a repeated measure design was used, in which all subjects (participants) were subjected to all four treatment groups. A washout period of 10 min was provided between each treatment/serving. In order to determine sight acceptability, mustard leaves were harvested from each treatment and given to the respondents for evaluation. Sight appearance was evaluated on a scale of 1 to 10, where 1 represented very bad looking and 10 represented excellent appearance (Table 1). For smell/odor acceptability, the participants rated the same mustard leaves for smell on a scale of 1 to 10, where 1 represented very smelly leaves and 10 excellent odor (Table 1). For evaluation of taste, mustard leaves were cleaned, cut, and cooked without the addition of any spices and condiments, except for salt to avoid taste-compounding factors. Then, each participant was served 10 g of the vegetable from each treatment for tasting and rated the taste on a scale of 1 to 10, where 1 represented very bitter and 10 represented excellent taste (Table 1). Each participant washed their mouth between each serving to avoid the influence of the previous serving on the next. The scores for the taste ranged from not bitter, less bitter, and most bitter respectively. The visual representations (Table 1) were additionally used for the participants to express their satisfaction with each serving. Participants were asked to circle the facial expression that corresponded to the level of satisfaction with each serving. This was especially useful for the less literate group of farmers who would have challenges with the numbered Likert scale. These data were used to complement the numbered Likert scale.

## 3. Results

### 3.1. Number of Affected Plants

In general, the number of affected plants with visible insect pest presence decreased as days from the day of application progressed (Table 2) in all plots sprayed with either plant extracts or synthetic pesticides. There were also significant differences observed (*p*_0.05_ < 0.001) in the number of affected plants during all 6 weeks of observation after the first and second applications. Among the treatments, the number of plants affected by insect pests was lowest in plots sprayed with synthetic pesticides followed by plots sprayed with neem. For example, after 1 week of the first application, 3.25, 3.50, and 4.25 plants were affected by pests in plots sprayed with synthetic, neem, and Gliricidia plant extracts, respectively. These differences were, however, not statistically significant but were significantly different from the untreated plot (7.50 plants). In all cases, however, the number of affected plants decreased over time by the third week after both the first and second application (Table 2).

### 3.2. Number of Insects

The key insects observed in the plots and the damage they caused are presented in Table 3. These were aphids, caterpillars, flea beetles, grasshoppers, and redbugs. Among these, aphids and flea beetles were the most abundant, especially in negative control plots (untreated/none) whereas caterpillars, grasshoppers, and redbugs were the least abundant. For example, about 30 aphids were observed in the untreated plots whereas none were observed in plots sprayed with both extracts and synthetic pesticide (Figure 2). However, among all the insects, the abundance was significantly higher (*p*_0.05_ < 0.001) in the untreated plots than the treated ones, whereas among the treated ones, the differences between plant extracts and synthetic insecticide were nonsignificant (*p*_0.05_ > 0.05, Figure 2). Notably, neem plant extracts reduced caterpillars and flea beetles’ abundance to a greater extent than synthetic insecticides, albeit the differences were insignificant.

### 3.3. Biological and Economic Yield

The biological yield was obtained to represent the net gain of photosynthates (dry matter) inclusive of all infested parts. In this regard, there were no significant differences observed in biological yield between treated and untreated plants (Figure 3A), suggesting that application of plant extracts and insecticides did not affect net photosynthate accumulation. However, economic yield was significantly different among the treatments (*p*_0.05_ < 0.001), with untreated (none) plants experiencing a significant reduction in economic yield, whereas no significant differences were observed between the treated plants—gliricidia extracts, neem extracts, and synthetic insecticides (Figure 3B).

### 3.4. Total Leaf Area

Total leaf area was obtained to evaluate the proportion of plant leaves usable for consumption after excluding infested leaves. Significant differences were shown in total leaf area between the treated and untreated plants (*p*_0.05_ < 0.001). Untreated plants had the lowest total leaf area (about 550 cm^2^ per plant) whereas plants sprayed with neem extracts had the highest total leaf area (about 680 cm^2^ per plant). Both plant extracts (neem and gliricidia) performed better in enhancing total leaf area than synthetic insecticides, albeit with the differences being statistically insignificant (Figure 4).

### 3.5. Beneficial Insects’ Abundance

In order to assess the ability of plant extracts to maintain beneficial insects, we assessed the abundance of beneficial insects in each plot. Five beneficial insects were identified and monitored during the experiment, these were beetles, butterflies, roaches, spiders, and wasps, which play various roles in the ecosystem (Table 4). Among these insects, roaches and wasps were the most abundant, whereas beetles were the least abundant (Figure 5). There were significant differences in insect abundance of all insects among the various treatments (*p*_0.05_ < 0.001). For all the insects, plots sprayed with synthetic insecticides had the most significantly lower beneficial insect abundance (Figure 5). Strikingly, this was followed by neem plant extracts, whereas the most abundant was in gliricidia plant extracts; this pattern holds true for beetles, spiders, and wasps (Figure 5). On the other hand, insect abundance for butterflies was higher in the untreated plots, whereas roaches were higher in neem plant extracts.

### 3.6. Membrane Integrity

Cognizant of the fact that insect damage may directly, through feeding, and indirectly through elicitation of reactive oxygen species (ROS) production, damage membranes, we performed an electrolyte leakage assay to determine membrane integrity. Here, it was shown that untreated plants suffered a greater extent of membrane damage as shown by significantly higher (*p*_0.05_ < 0.001) electrolyte leakage (Figure 6), whereas no differences were observed between plant extracts and synthetic insecticide.

### 3.7. Sensory Attributes

In order to investigate whether the application of plant extracts influenced the sensory attributes of leaf mustard, a sensory evaluation was conducted. Three sensory attributes, namely, taste, appearance, and odor, were evaluated. Based on the physical appearance of mustard leaves, significant differences (*p*_0.05_ = 032) were observed among the treatments, with untreated plants producing the worst-looking leaves. Synthetic insecticides and neem plant extracts produced the best-looking leaves (Table 5), which were considered excellent according to the scale shown in Table 2. In terms of taste, there were significant differences (*p*_0.05_ = 0.023) in the taste of leaf mustard sprayed with different treatments. Synthetic insecticides produced the most bitter leaves followed by neem, whereas gliricidia produced the least bitter leaves and were not significantly different from untreated plants (Table 5). Furthermore, significant differences (*p*_0.05_ = 0.038) were shown in the odor produced from the mustard leaves after harvest (Table 5). Synthetic insecticides produced the smelliest leaves. Strikingly, mustard plants applied with gliricidia plant extracts produced fewer smelly leaves than untreated plants, albeit the differences were insignificant from both untreated and neem plant extracts (Table 5).

### 3.8. Discussion

Agricultural production is increasingly being confronted with a wide array of biotic and abiotic constraints that have placed enormous pressure on sustainably maintaining productivity to meet growing food demand. Pest management is among the central hurdles that farmers must regularly deal with and has inevitably necessitated unwarranted utilization and over-reliance on synthetic pesticides. This has resulted in an outcry regarding the economic and environmental implications, including non-target toxicity, residual consequence, and challenging biodegradability, requiring an alternative and prompt adoption of sustainable and cost-effective pest control measures [15]. In this study, we explored the potential of neem and gliricidia botanical extracts in managing pest infestation in leaf mustard. In the study, it was found that botanical extracts of neem and gliricidia plants were able to significantly reduce insect pest abundance in leaf mustard as well as enhance economic yield and quality (Table 2, Figure 2, Figure 3, Figure 4 and Figure 5). For example, both botanicals were able to completely prevent aphid accumulation and significantly reduce caterpillars (Figure 2). Of the two botanicals, neem extracts had the highest efficacy in managing insect pest prevalence, the number of affected plants, and minimizing leaf membrane integrity (Table 2, Figure 2 and Figure 6). Neem botanical pesticides have been utilized to manage insect pests, including aphids, armyworms, bean leaf spot, bollworms, cabbage loopers, caterpillars, common grasshoppers, bruchid beetle, pink stalk borer, and thrips [16].

Consistent with previous studies, this study has demonstrated the insecticidal efficacy of neem botanical extracts on aphids, grasshoppers, redbugs, flea beetles, and caterpillars (Figure 2). Its insecticidal properties have been implicated in the presence of azadirachtin, an active ingredient that exhibits multiple modes of action, including anti-feedancy, detrimental effects on morphology, alteration in biological fitness, fecundity suppression, decreased growth, oviposition repulsion, and even sterilization [17]. The study also found a higher efficacy of gliricidia botanical extracts that significantly reduced insect pest abundance and enhanced leaf mustard yield and quality. *Gliricidia sepium* is one of the most promising plants with insecticidal properties [9,18,19,20]. A study by Montes-Molina et al. (2008) [18] evaluated the efficacy of neem and gliricidia botanical extracts on controlling pests in maize in comparison with untreated (control) and synthetic pesticides, in which, similar to this study, gliricidia and neem extracts showed greater efficacy in reducing insect pest damage to new leaves and increased maize yield. In another study, neem plant extracts have also been reported to show 100% mortality against storage flour beetles (*Tribolium confusum*) [21]. Jose and Sujatha (2017) [22] demonstrated the antifeedant activity of *Gliricidia sepium* extracts against third-instar larvae of *Helicoverpa armigera* (Hubner) (Lepidoptera: Noctuidae). Neem and gliricidia leaf extracts were also evaluated in a study by Montes-Molina et al. (2014) [23] as potential insect repellants under organic tomato cultivation, and it was found that *G. sepium* stimulated tomato leaf and fruit characteristics. This was primarily ascribed to its role in growth regulation, and not as an insect repellant. Hence, the better growth characteristics of plants sprayed with *G. sepium* in this study could in part be attributed to its better growth regulation properties, in addition to its insecticidal properties.

A key benefit of botanical extracts over synthetic pesticides is their ability to preserve some beneficial insects, a central principle in integrated pest management (IPM) approaches cognizant that not all “insects” are “pests”; this has been adequately demonstrated. For example, Tembo et al. (2018) [5] found that pesticidal plant extracts improved yield and reduced insect pests on legume crops without harming beneficial arthropods in common beans, cowpeas, and pigeon peas. Similar findings have been reported in Montes-Molina et al. (2008) [18], where neem and gliricidia plant extracts significantly preserved beneficial insect abundance in maize by over 100% relative to synthetic pesticides and comparable to untreated plots. In this study, we have shown that beneficial insects were significantly preserved in leaf mustard by both gliricidia and neem extracts (Figure 5) by levels nearly comparable with untreated plots and significantly higher than synthetic pesticides. However, a prominent observation was that, while neem extracts had significantly higher beneficial insect abundance than synthetic pesticides, it was significantly lower than gliricidia, which exhibited a beneficial insect abundance higher than untreated plants for spiders and wasps. This is indicative that gliricidia extracts are more effective at preserving beneficial insects and maintaining ecosystem services. These beneficial insects play various ecosystem roles such as predation of harmful insect pests, pollination, and conversion of wastes into nutrients (Table 3). Hence, the findings of this study demonstrate that the utilization of botanical extracts of pesticidal plants to control pests can be as effective as synthetic insecticides in terms of crop yields while also preserving tritrophic interactions between plants, insect pests, and their natural enemies, conserving the non-target insects that provide important ecosystem services such as pollination and pest regulation [5]. Thus, managing crop pests using botanical extracts can be more easily integrated into agro-ecologically sustainable crop production systems.

One of the harmful effects of insect pests on plants physiologically is membrane damage inflicted directly through feeding and indirectly through elicitation of reactive oxygen species (ROS). Thus, membrane integrity assays have been widely utilized to evaluate plants responses under biotic and abiotic stresses [13,24,25,26,27]. The electrolyte leakage assay is one of the widely utilized techniques to study cellular membrane integrity. In this study, insect pest damage inflicted considerable membrane damage as shown by higher electrolyte leakage in untreated plants (Figure 6). Strikingly, plants sprayed with botanical extracts experienced very minimal electrolyte leakage comparable to plants sprayed with synthetic insecticides (Figure 6), indicative that neem and gliricidia extracts effectively minimized membrane damage. However, available data cannot ascertain whether the reduction is ascribed to a reduction in ROS generation, but reduction through reduced direct feeding is certainly the most plausible explanation.

A major setback often cited to limit the adoption of organic farming techniques is the unsatisfactory quality, particularly relating to the appearance of organically grown food produce [28,29]. This is in sharp contrast with other sensory attributes such as flavor, where it has been claimed that organic growing methods produce more flavorful and better-tasting fruits and vegetables [30]. Sensory attributes such as appearance, taste, and odor are critical determinants in influencing decisions to purchase and consume a commodity, but these are often neglected in studies evaluating botanical pesticides as central emphasis has been placed on insecticidal efficacy evaluation. Here, we show significant differences in sensory attributes in leaf mustard in all three parameters investigated. Our findings suggest that neem botanical extracts produce relatively bitter leaves than gliricidia extracts, nearly comparable to synthetic insecticides. However, the visual quality of their (neem extract) leaves is far greater than gliricidia and likewise, comparable to synthetic insecticides. These findings, in part, agree with the sentiments discussed by Bourn and Prescott (2002) [30]. Plausibly, the relatively bitter taste from neem extracts may be emanating from its richer bioactive compounds which may also have contributed to its strong odor. Nonetheless, their insecticidal efficacy offset their relatively weaker associated sensory attributes. In the extracts used in the present study, based on the combined effects of reduced pest infestation, plant damage and sensory attributes, and maintenance of beneficial insects, gliricidia extracts would be the ultimate recommendation, but based on insecticidal properties and visual quality of the produce, neem extracts are a better candidate. Hence, efforts may have to be intensified towards optimizing their formulation to enhance sensory properties and beneficial insects’ preservation.

Insect pest infestation is one of the major challenges faced in leafy vegetable crop production attracting unwarranted use of synthetic pesticides. Findings from this study, therefore, provide much-needed evidence and optimism that using botanical extracts for crop production can support the farmers’ transition from chemical-intensive farming techniques towards more sustainable agro-ecological approaches to crop production. Considering the growing concerns about pesticide residues in foods, environmental pollution, pesticide resistance development in insect pests, and health hazards to farmers as well as the growing interest in organically grown food, the development of sustainable organic production systems for vegetable crops is essential for consumer satisfaction and grower competitiveness. Hence, more research is imperative, such as conducting field trials under both smallholder farmers’ conditions and commercial farming enterprises, to evaluate the feasibility of using pesticidal plants for smallholder and commercial crop production in order to generate more evidence for their integration into agro-ecologically sustainable vegetable crop production systems.

## Figures and Tables

**Figure 1 insects-16-00156-f001:**
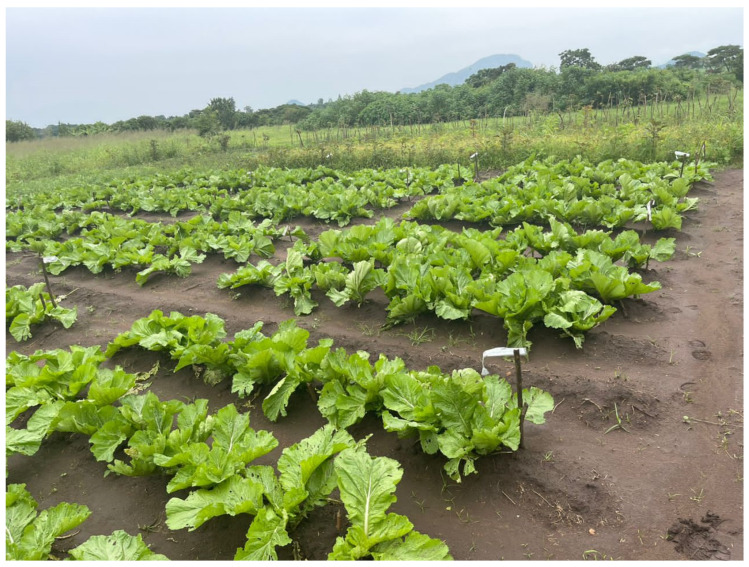
Leaf mustard plots with 4 blocks in a randomized complete block design.

**Figure 2 insects-16-00156-f002:**
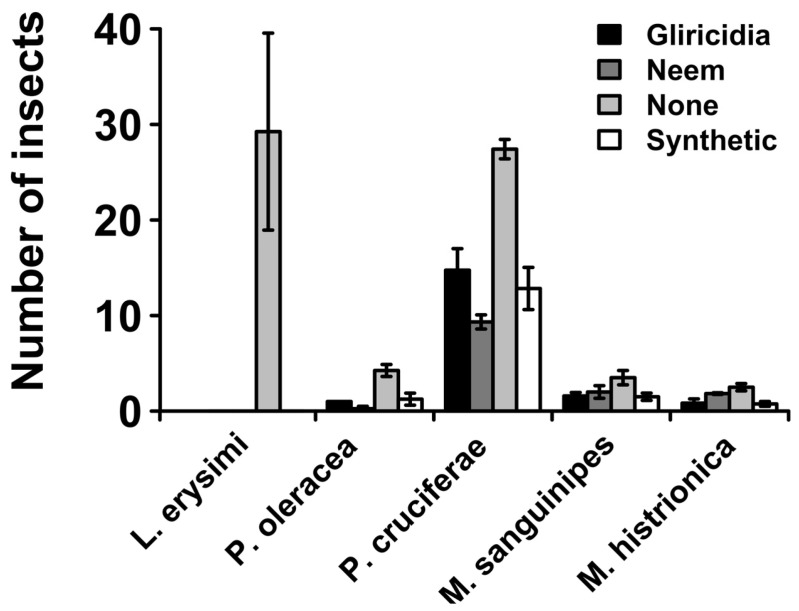
Average number of insect pests in treated and untreated plots per scouting week. The values represent weekly means from 4 plots in the 4 blocks.

**Figure 3 insects-16-00156-f003:**
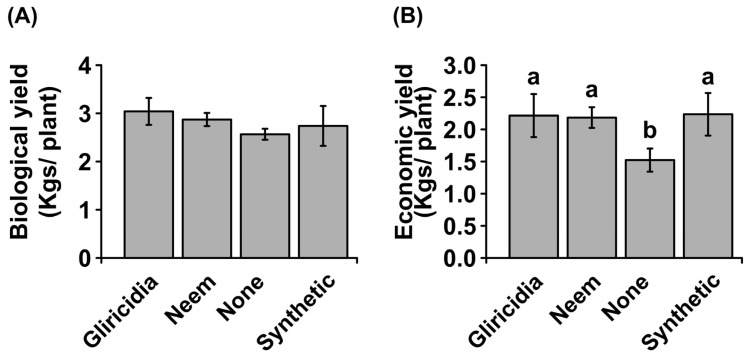
Effect of botanical extracts and synthetic insecticide on biological (**A**) and economic yield (**B**) of mustard leaves. The values represent means from 10 biological replicates. Different letters indicate significant differences using the Tukey test at a 0.05 level of significance, whereas similar letters indicate no significant differences at a 0.05 level of significance.

**Figure 4 insects-16-00156-f004:**
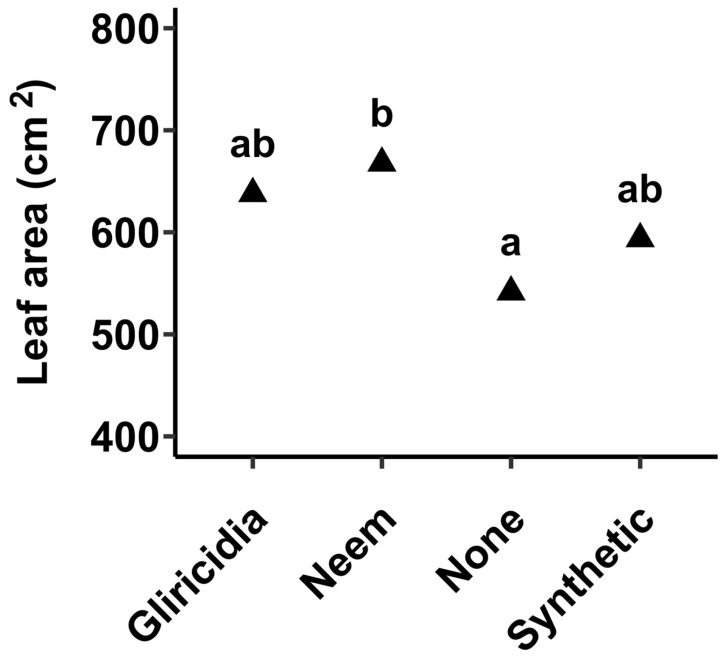
Effect of botanical extracts and synthetic insecticide on economic leaf area of mustard leaves. The values represent means from 10 biological replicates. Different letters indicate significant differences using the Tukey test at a 0.05 level of significance, whereas similar letters indicate no significant differences at a 0.05 level of significance.

**Figure 5 insects-16-00156-f005:**
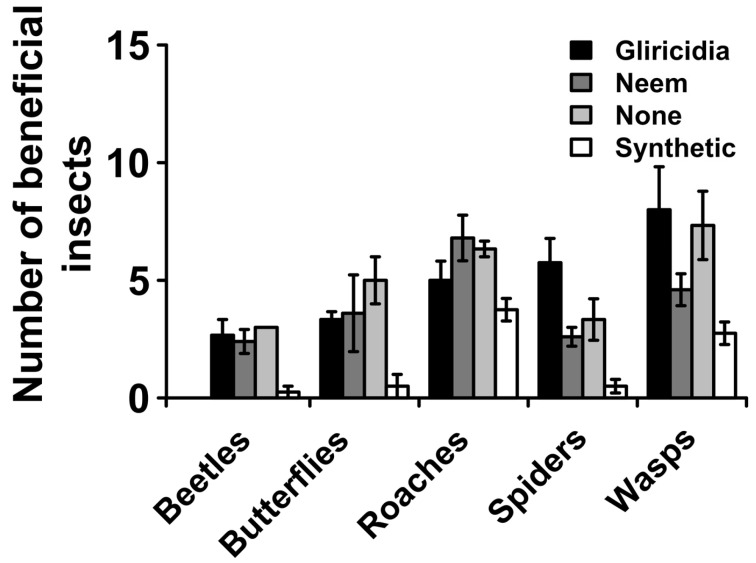
Effect of plant botanical extracts and synthetic insecticides on beneficial insect abundance. The values represent means from 4 plots in the 4 blocks.

**Figure 6 insects-16-00156-f006:**
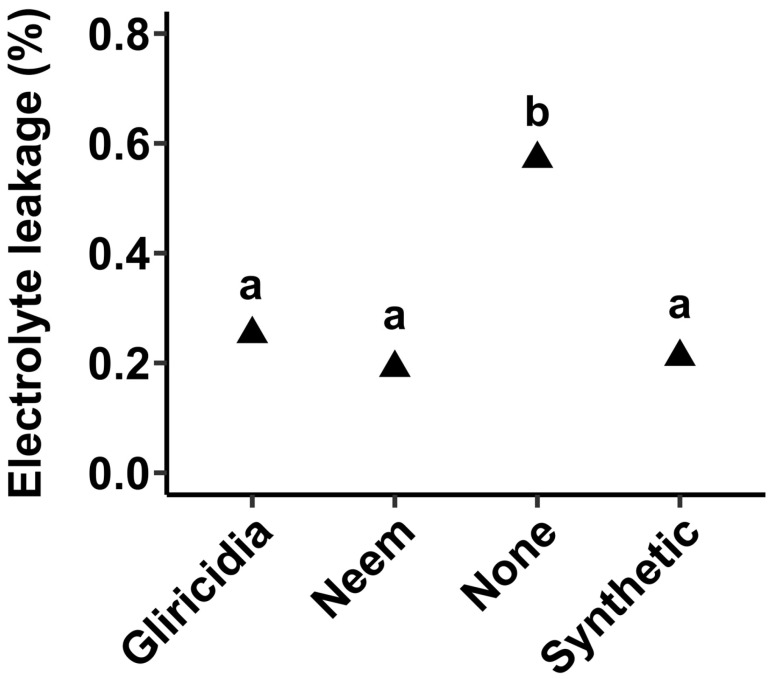
Effect of plant botanical extracts and synthetic insecticides on membrane integrity. The values represent means from 10 biological replicates. Different letters indicate significant differences using the Tukey test at a 0.05 level of significance, whereas similar letters indicate no significant differences at a 0.05 level of significance.

**Table 1 insects-16-00156-t001:** Scoring method used for sensory attributes.

Scores	Visual	Taste	Appearance	Odor
1 to 2	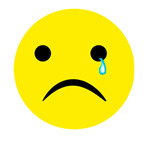	Very bitter	Very bad	Very smelly
3 to 4	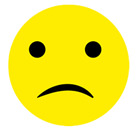	Bitter	Bad	Smelly
5 to 6	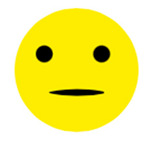	Neutral	Neutral	Neutral
7 to 8	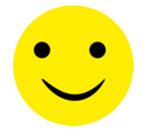	Not so bitter	Good	Good
9 to 10	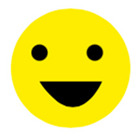	Excellent taste	Excellent	Excellent

**Table 2 insects-16-00156-t002:** Effects of pesticides on the number of affected plants. Means with different letters indicate significant differences using the Tukey test at a 0.05 level of significance, whereas similar letters indicate no significant differences at a 0.05 level of significance.

Treatment	Week 1	Week 2	Week 3	Week 1	Week 2	Week 3
Synthetic	3.25 ^a^	2.75 ^a^	1.50 ^a^	3.25 ^a^	3.50 ^a^	1.50 ^a^
Neem	3.50 ^a^	3.50 ^a^	2.00 ^a^	4.25 ^a^	3.75 ^a^	3.25 ^b^
Gliricidia	4.25 ^a^	4.25 ^ab^	2.75 ^a^	4.50 ^a^	4.00 ^a^	2.00 ^ab^
Untreated	7.50 ^b^	7.50 ^c^	7.25 ^b^	6.25 ^b^	7.75 ^h^	7.50 ^c^
F_0.05_	<0.001	<0.012	<0.001	<0.001	<0.001	<0.001

**Table 3 insects-16-00156-t003:** A description of insect pest species identified in the leaf mustard field.

Latin Name	Insect Order	Common Name	Damage to the Plant
*Lipaphis erysimi* (Kalt)	Hemiptera	Mustard aphid	Nymphs and adults suck sap from leaves and cause damage by poor plant growth and curling of leaves.
*Pieris oleracea* (Harris)	Lepidoptera	Caterpillar	Young larvae feed gregariously mostly on the undersurface of the leaves. Caterpillars feed on leaves and in severe infestation the whole crop is defoliated.
*Phyllotreta cruciferae (Goez)*	Coleoptera	Flea beetle	Adult beetles feed on the surface of leaves and stems and produce small pits. The tissue underneath the injury eventually withers and dies.
*Melanoplus sanguinipes*	Orthoptera	Grasshopper	Directly feed on the leaves causing defoliation.
*Murgantia histrionica (Hahn)*	Hemiptera	Redbug	Damage to the foliage as they feed with their piercing/sucking mouthparts.

**Table 4 insects-16-00156-t004:** Beneficial arthropods identified and their ecosystem roles.

Latin Name	Insect Order	Common Name	Ecosystem Service
*Coccinellidae* spp.	Coleoptera	Lady beetle	Beetles and larvae are predators of aphids, mites, insect eggs, small insect larvae, and hoppers.
*Trichogramma minutum*	Lepidoptera	Parasitic wasp	Parasitizes borer larvae, stem-borer moth, and earworm moths’ eggs.
*Araneae* spp.	Araneae	Spiders	Predators for a range of insect pests.
*Lepidoptera* spp.	Lepidoptera	Butterfly	Pollination.
*Blattodea* spp.	Blattodea	Roaches	Convert waste into nutrients for soil and earth.

**Table 5 insects-16-00156-t005:** Effects of plant extracts and synthetic insecticides application on sensory attributes of leaf mustard. Means with different letters indicate significant differences using the Tukey test at a 0.05 level of significance, whereas similar letters indicate no significant differences at a 0.05 level of significance.

Treatments	Taste	Appearance	Odor
Gliricidia	4.5 ^b^	6.9 ^b^	3.9 ^b^
None	3.5 ^b^	4.3 ^a^	3.5 ^b^
Neem	2.5 ^ab^	8.0 ^bc^	2.7 ^b^
Synthetic	2.0 ^a^	9.0 ^c^	1.0 ^a^
*p* _0.05_	0.023	0.032	0.038
CV (%)	29.40%	21.40%	28.10%

## Data Availability

The original contributions presented in the study are included in the article, further inquiries can be directed to the corresponding author.

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
