# Peer review of "Neem and Gliricidia Plant Leaf Extracts Improve Yield and Quality of Leaf Mustard by Managing Insect Pests’ Abundance Without Harming Beneficial Insects and Some Sensory Attributes"

_insects, 2025, doi:10.3390/insects16020156_

Round 1
Reviewer 1 Report
Comments and Suggestions for Authors
This study is original as it addresses the efficacy of two plant-based insecticides (Neem and Gliricidia) on several pests together affecting mustard leaves, which has not been explored or has been scarcely explored, especially under field conditions, and even mor so considering the conditions of Malawi.
This study is the first to address these plant-based insecticides on multiple pest species together under these field conditions.
In the title it suggests eliminating the term "harmful" since the term "Pest" implies it.
I also suggest including the scientific names of the pests studied in the summary.
In the method the authors should specify how they identified pest species (Line 133) and natural enemies (Lines 139-140)?, e.g. identification keys, by comparison, etc. They should also detail how they made the weekly observations, e.g. how much time they spent observing each plant? How the spraying was done?, e.g. knapsack sprayer? (Lines 128-135).
In discussion,
I suggest discussing the effectiveness of these extracts (Neem or Gliricidia) on some of the pest species studied with other laboratory studies or on other crops.
The conclusions from the results are pertinent and consistent and address the main question since they address, that is, the efficacy of the tested botanical products and their eventual application in mustard crops.
References are appropriate but could be improved by incorporating more specific bibliography on the effectiveness of the pests studied in the discussion of the results.
Tables and figure are appropriate for presenting the results, but I suggest in Figure 2, using the scientific names of the pests instead of the common ones in order to highlight them.
Comments on the Quality of English Language
Moderate editing of English language required.
Reviewer 2 Report
Comments and Suggestions for Authors
Review report:
The study presented in this manuscript investigates the potential of neem and gliricidia botanical extracts as sustainable alternatives to synthetic insecticides for pest management in leaf mustard crops. The authors highlight the growing demand for vegetable crops and the consequent increase in the use of synthetic chemicals, which negatively impact human health and the environment. The study's findings suggest that neem and gliricidia extracts effectively manage harmful insect pests while preserving beneficial insects and maintaining crop quality.
The topic is highly relevant, considering the global shift towards sustainable agriculture and the increasing demand for safer, environmentally friendly pest management solutions. The study addresses an important issue by exploring alternatives to synthetic insecticides. The experimental approach is well-structured, with appropriate controls in place to compare the efficacy of botanical extracts against synthetic insecticides. The inclusion of multiple assessment parameters, such as pest prevalence, beneficial insect preservation, crop yield, sensory attributes, and physiological impacts, provides a comprehensive evaluation of the treatments. The study reports promising results, indicating that neem and gliricidia extracts are effective in reducing harmful insect pest populations while preserving beneficial insects. Additionally, the extracts did not adversely affect the sensory attributes of the leaf mustard, which is crucial for consumer acceptance. The findings emphasize the potential of botanical extracts to serve as environmentally friendly pest management solutions, aligning with the growing demand for sustainable agricultural practices.
Comments to authors
The Materials and Methods section of the manuscript provides a detailed description of the study's experimental design, location, and methodologies. However, there are several shortcomings that need to be addressed to improve the clarity, rigor, and reproducibility of the study:
The description of the study site lacks critical details about the soil type, vegetation cover, and other environmental factors (e.g., humidity, wind patterns) that could influence the results. These details are essential for understanding the context of the study and for replicating the experiment in other locations.
The preparation of plant extracts mentions the use of a "water extraction method" but does not provide sufficient detail on the steps involved in the extraction process, such as the duration and conditions (e.g., temperature) of the extraction, and whether the extracts were standardized for bioactive compound concentration. Additionally, the manuscript does not mention if there were any controls to account for potential variations in extraction efficiency.
The rationale behind choosing specific methods (e.g., 10% w/v concentration for plant extracts, 4-day interval for pesticide application) is not well justified. It would be beneficial to include references to prior studies or preliminary experiments that support these choices.
Some procedures, such as the sensory evaluation, lack precise details on how the tests were administered. For example, it is unclear how the participants were selected, how the visual representations were adapted from the cited study, and whether any bias control methods (e.g., randomization, blinding) were implemented.
The method for observing and recording insect pest populations and beneficial insects is vaguely described. There is no mention of the frequency, duration, or specific techniques used for monitoring insect populations (e.g., visual counts, traps). Furthermore, the criteria for identifying and classifying insects as pests or beneficial are not clearly outlined.
Given that the field trials were conducted under open-field conditions, it is crucial to account for environmental variability (e.g., microclimate differences, pest pressure) across the plots. The manuscript does not mention any steps taken to control or account for this variability.
By addressing these shortcomings, the authors can enhance the methodological rigor of the study, making it more robust and easier for other researchers to replicate and build upon their work.
I recommend major review.
Round 2
Reviewer 1 Report
Comments and Suggestions for Authors The manuscript was significantly improved and is ready to be published inthis journal.
Reviewer 2 Report
Comments and Suggestions for Authors
The authors have addressed all my questions. I recommend that the article be accepted!